# Progress in Anticancer Drug Development Targeting Ubiquitination-Related Factors

**DOI:** 10.3390/ijms232315104

**Published:** 2022-12-01

**Authors:** Qianqian Li, Weiwei Zhang

**Affiliations:** College of Life Sciences, Capital Normal University, Beijing 100048, China

**Keywords:** ubiquitination, cancer, drug target

## Abstract

Ubiquitination is extensively involved in critical signaling pathways through monitoring protein stability, subcellular localization, and activity. Dysregulation of this process results in severe diseases including malignant cancers. To develop drugs targeting ubiquitination-related factors is a hotspot in research to realize better therapy of human diseases. Ubiquitination comprises three successive reactions mediated by Ub-activating enzyme E1, Ub-conjugating enzyme E2, and Ub ligase E3. As expected, multiple ubiquitination enzymes have been highlighted as targets for anticancer drug development due to their dominant effect on tumorigenesis and cancer progression. In this review, we discuss recent progresses in anticancer drug development targeting enzymatic machinery components.

## 1. Mechanisms of Ubiquitination

In the process of ubiquitination, ubiquitin (Ub) protein is covalently attached to substrate proteins either as a monomer (monoubiquitination) or as a polymer chain (polyubiquitination). Monoubiquitination simply adds a single Ub to one or multiple lysine (K) residues of substrates. Polyubiquitination modifies proteins with the C-terminal glycine residue of the Ub (Ub-G76), which is linked with one K residue of the proximal Ub via an isopeptide bond. All K residues in Ub, including K6, K11, K27, K29, K33, K48, and K63, can serve as a linkage site. Of note, in some particular cases, Ub can be conjugated to non-K residues, such as methionine-1 (M1), cysteine (C), serine (S), threonine (T), and tyrosine (Y) [1,2,3,4]. A poly-Ub chain is generally homogenous where all the Ub units are linked through the same residue sites. Of note, atypical types of poly-Ub chains are observed in eukaryotic cells [5,6], including mixed and branched poly-Ub chains. In mixed poly-Ub chains, the Ub units employ distinct residues for linkage [7], while branched poly-Ub chains contain a Ub unit providing more than one K residue concurrently for linkage formation (Figure 1) [1,2,3,4]. Upon ubiquitination, protein stability or activity is modified, and the linkage types largely decide the fate of substrates [8]. In general, mono-Ub, K6-, K11-, K29-, and K48-linked poly-Ub chains generally act as a “death label” for proteins and drive them for proteasome or lysosome-mediated degradation [9,10]. K63-linked poly-Ub chains usually regulate the activity and subcellular localization of substrate proteins [11,12,13]. Interestingly, the same Ub linkage could result in distinct outcomes. For instance, the K63-linked poly-Ub chain plays a role of a degradation inducer for octamer-binding transcription factor-4 (Oct4) in mouse embryonic stem cells [14,15,16]. The K11-linked poly-Ub chain does not serve as a “death label” for β-catenin but increases its stability in human colorectal cancer cells [17,18].

Ubiquitination is achieved by three types of enzymes, including the Ub activating enzyme E1, Ub binding enzyme E2, and Ub ligase E3. In this process, E1 produces the Ub–adenosine monophosphate (Ub–AMP) intermediate through ATP-dependent adenylation at Ub-G76. Next, the Ub–AMP attacks the E1 active C site via nucleophilic reaction. A thioester bond forms between the E1 and Ub-G76 accompanying AMP release [19]. Subsequently, E2 mediates a transthiolation reaction to accept Ub at its active C site. At the final step, an E3 ligase simultaneously associates with the E2-Ub and a protein substrate, mediating isopeptide bond formation between the Ub and the substrate [8,20,21]. During this process, E2s largely decide the length and topological structure of Ub chains, while E3s usually specifically recognize substrates through protein–protein interactions [20,22]. In human cells, there are two E1s, 40 E2s, and over 600 E3s [23,24,25]. All E2s are characterized by a ~150 amino acid enzymatic UBC domain with the active site C. Those E2s only possessing the UBC domain in sequence are grouped as class I, while class II and III E2s additionally possess an extended C- and N-terminus, respectively. Class IV E2s contain extension regions at both ends. The extension regions usually regulate subcellular localization of E2s and mediate their interaction with other partners [25,26]. E3s can be divided into three types based on their sequences and Ub transferring mechanisms, including the really interesting new gene (RING)-type, homologous to the E6-AP COOH terminus (HECT)-type and RING between RING (RBR)-type E3s [27]. The RING-type E3s contain a characteristic RING or U-box domain with a similar RING finger fold in the structure. In the process of Ub transfer, the RING-type E3s directly hand over Ub from the E2s to the substrates [27], during which the RING domain, rather than the U-box, displays zinc ions (Zn^2+^-dependent). To date, more than 500 RING-type E3s have been identified [27]. There are nearly 30 HECT-type E3s encoded in human cells [28]. In addition to the catalytic HECT domain, these E3s usually contain a variable N-terminal domain responsible for substrate recognition [29]. According to the sequences of the N terminus, HECT-type E3s can be further divided into three groups, including the WW domain-containing neuronal precursor cell-expressed developmentally downregulated 4 (Nedd4)/Nedd4-like E3s, HERC (HECT and RCC1-like)- and RLD (RCC1-Like domains)-containing E3s, and the HECT-type E3s without WW and RLD domains [30]. During ubiquitination, HECT E3s achieve Ub transfer through a two-step reaction. Firstly, Ub is transferred from E2 to the catalytic C site of E3 to form a HECT–Ub thioester intermediate. Next, the Ub is ligated to the substrates [27]. In human cells, there are 14 RBR-type E3s encoded, all of which are featured by two conserved Zn^2+^-binding RING domains, RING1 and RING2 [31]. RBR-type E3s employ similar two-step reactions loading Ubs to substrates despite the absence of HECT domains, which attributes to the catalytic active site of C in RING2 [32].

Ubiquitination is widely involved in various biological processes, such as ribosome function [33], DNA replication [34,35], transcription [36], and inflammatory response [37]. Abnormal ubiquitination of key proteins could cause severe illnesses, such as metabolic disorder, inflammation, degenerative diseases, and malignant cancers [38]. Thus, targeting the ubiquitination process for drug development to achieve improved disease therapy, especially in the anticancer field, has become a research hotspot in recent years.

## 2. Cancer-Related Ubiquitination Factors

### 2.1. E1

The ubiquitin proteasome system (UPS) is an important process regulating protein homeostasis. Inhibition of E1 UAE can cause the accumulation of misfolded proteins in the endoplasmic reticulum, thereby destroying the homeostasis of the endoplasmic reticulum and allowing cells to enter the process of apoptosis [39,40]. This strategy appears effective to treat B-cell lymphoma and malignant hematological diseases (Table 1) [39,40]. Furthermore, UAE inhibition can stabilize some tumor suppressors, such as p53, thus delaying the progress of cancer development [41,42]. There is also evidence showing that this inhibition activates the NF-κB signaling pathway so as to suppress malignant tumor growth [43].

### 2.2. E2-Ubiquitin-Conjugating Enzyme 2C (E2-Ube2c)

*Ube2c* gene is located on chromosome 20q13.12 encoding a 179 amino acid E2 with a molecular weight of 19 kDa. It cooperates with the anaphase promoting complex/cyclome (APC/C) E3 complex to polyubiquitinate proteins for degradation [44,45]. The activity of Ube2c (Table 1) is involved in cell-cycle regulation and loss of genetic stability [46]. In cancer cells, Ube2c inhibits apoptosis and enhances tumor growth [47]. Ube2c overexpression has been detected in various cancers, such as hepatocellular carcinoma [48], esophageal cancer [49,50], and breast cancers [51,52]. In gastric cancer cells, Ube2c inhibition interrupts cell-cycle progression, while its elevation enhances cancer cell proliferation [47]. In rectal cancers, siRNA-mediated *Ube2c* depletion induces apoptosis and inhibits cell proliferation, further disturbing cancer growth and invasion [44].

### 2.3. E3s

#### 2.3.1. Glycoprotein 78 (Gp78)

*Gp78* is evolutionarily conserved and encodes a transmembrane glycoprotein possessing the leucine zipper motif and RING domain mediating E3 ligase activity [53,54]. Gp78 is localized on the endoplasmic reticulum membrane and interacts with endoplasmic reticulum-associated degradation (ERAD) E2s via the C terminus of its RING domain [55,56]. Gp78 promotes tumor metastasis and proliferation [57]. In sarcomas, Gp78 interacts with the transmembrane metastasis inhibitor KAI1 (also known as CD82), and inhibition of Gp78 can promote the accumulation of KAI1, lead to apoptosis, and reduce tumor metastasis (Table 1) [58].

#### 2.3.2. Mouse Double Minute 2/Human Homolog of Mouse Double Minute 2 (MDM2/HDM2)

Tumor protein p53 (TP53) (also known as p53) is a tumor suppressor widely involved in critical signaling pathways, including autophagy, aging, differentiation, proliferation, DNA repair, and tumorigenesis [59,60,61,62,63,64]. p53 dysfunction is closely related to human cancers and neurodegenerative diseases [65,66]. Deletion in the *p53* gene aggravates deterioration of cancers [67,68,69,70]. RING-type E3 MDM2 binds to p53 through its N terminus to ubiquitinate p53 protein [71,72,73]. MDM2 mediates multiple monoubiquitination of p53 to promote its nuclear export to cytoplasm for proteasomal degradation [74]. MDM2 can also modify p53 via poly-Ub chain for degradation in the nucleus [75,76]. Furthermore, MDM2 binds to the transactivation domain of p53 and inhibits its transcriptional activity [77]. The activity of MDM2 in repressing p53 determines its oncogenetic effect. It has been shown that MDM2 displays an elevated expression in various human cancers, such as osteosarcoma, neuroblastoma, lung cancer, colon cancer, breast cancer, and liver cancer, where it enhances p53 degradation, leading to poor survival and prognosis of patients (Table 1) [78,79,80]. In fact, in addition to MDM2, multiple RING-type E3s are involved in regulating p53 levels, including constitutive photomorphogenesis protein 1 (COP1), p53-induced protein with a RING-H2 domain (Pirh2), and co-chaperone carboxyl terminus Hsp70/90-interacting protein (CHIP). These E3s ubiquitinate p53 to inhibit its function and, thus, promote cancer development [81].

#### 2.3.3. SCF^Skp2^ Complex

The RING-type E3 complex Skp1–Cul1–Skp2 (SCF^Skp2^) is composed of multiple subunits including F-box protein S-phase kinase-associated protein 2 (Skp2), Cullin 1 (Cul1), Ring box 1 (Rbx1, also known as Roc1), and Skp1. Cul1 serves as a rigid scaffold to allow assembly of Rbx1 and Skp1 [82,83]. Rbx1 recruits E2 through its RING domain and constitutes the catalytic center with Cul1 to allow direct transfer of Ub to substrates [82]. Skp1 provides a docking site for Skp2. Skp2 specifically recognizes substrates via its F-box domain [84]. Skp2 (Table 1) is oncogenetic. Deficiency in Skp2 inhibits the development of breast cancer [85]. Several substrates have been identified to mediate its roles in tumorigenesis and malignancy. For instance, it ubiquitinates serine/threonine kinase Akt with K63-linked poly-Ub chain in the process of tumorigenesis. Cancers can also occur with enhanced degradation of forkhead box O1 (FOXO1) by Skp2 [86,87]. Transcriptional coactivator yes-associated protein (Yap) promotes tumor growth and metastasis [88]. Deletion of Yap can inhibit the development of breast cancer [89]. Skp2 can promote its activity through ubiquitination with K63-linked poly-Ub chain [90]. MutT homolog-1 (MTH1) is a nucleotide pyrophosphatase that oxidizes dNTPs to protect genomic DNA and prevents reactive oxygen species (ROS)-induced cytotoxicity in cancers [91,92]. In melanoma cells, MTH1 can be regulated by Skp2-mediated K63 polyubiquitination, which stabilizes MTH1 protecting DNA integrity [93].

#### 2.3.4. Inhibitor of Apoptosis-Related Proteins (IAPs)

IAPs are RING-type E3s [94]. The human genome encodes five IAPs proteins in total, which are X-linked inhibitor of apoptosis (XIAP), cellular inhibitor of apoptosis proteins 1 and 2 (cIAP1/2), melanoma-inhibitor of apoptosis (ML-IAP), and inhibitor of apoptosis-like protein-2 (ILP2). In addition to catalytic C-terminal RING domains, their characteristic sequence is the baculoviral IAP repeat (BIR) responsible for protein–protein interaction [95,96,97,98]. IAPs are involved in regulating diverse biological processes, including apoptosis, immune response, cell cycle, and migration [99,100]. Many types of human cancers display overexpressed IAPs that enhance tumor growth. For instance, IAPs (Table 1) repress apoptosis of melanoma cells for uncontrollable growth [101]. In contrast, there is evidence showing a putative role of IAPs as a tumor suppressor since cIAP deficiency constitutively activates NF-κB signaling in malignant hematopoietic cancer cells [102].

#### 2.3.5. WW Domain-Containing E3 Ubiquitin Protein Ligase 2 (Wwp2)

Wwp2 (also known as AIP-2, atrophin-1-interacting protein 2) is an HECT-type Nedd4 family E3 [103]. In addition to the HECT domain at the C terminus, Wwp2 contains a C2 domain at the N terminus and a tryptophan–tryptophan (WW) domain in the middle [104]. The WW region specifically recognizes phosphorylated PPxY motif-containing substrates [105]. One of the dominant substrates of Wwp2 is tumor suppressor phosphatase and tensin homolog (PTEN). PTEN is a lipid phosphatase [106]. It can inhibit tumor occurrence by repressing the PI3K/Akt pathway, and it plays an important role in cell survival and apoptosis regulation [107]. Mutations in PTEN can be observed in breast/prostate cancers and other human diseases [108,109,110]. Wwp2 can bind with the phosphatase domain of PTEN to promote its ubiquitination and degradation. In addition, another HECT-type E3 neural precursor cell expressed developmentally downregulated 4-1 (Nedd4-1) is involved in mediating PTEN polyubiquitination for degradation (Table 1) [104,110,111].

**Table 1 ijms-23-15104-t001:** Ubiquitination factors involved in cancer regulation.

Ubiquitination Factors	Cancer Types	References
UAE	B-cell lymphoma and malignant hematological diseases	[39,40,43]
Ube2c	Hepatocellular carcinoma, esophageal cancer, breast cancers, and gastric cancer	[47,48,49,50,51,52]
Gp78	Sarcomas	[58]
MDM2/HDM2	Osteosarcoma, neuroblastoma, lung cancer, colon cancer, breast cancer, and liver cancer	[78,79,80]
SCF^Skp2^ complex	Breast cancer and melanoma cells	[85,86,87,88,89]
IAPs	Melanoma cells and malignant hematopoietic cancer	[101,102]
Wwp2	Breast cancer and prostate cancer	[108,109,110]

## 3. Drug Development Targeting Ubiquitination-Related Factors

Since ubiquitination largely determines the stability and activity of proteins, it could be effective to develop drugs specifically targeting this process so as to achieve a better therapeutic effect. This has been become a hotspot in the field of disease therapy. These drugs can be derived from small molecules, peptides/proteins, or oligonucleotides [112]. The strategy to identify candidate targets for drug development requires combining computer-aided drug design methods and a high-throughput dataset of disease-related transcriptomes, drug responses, protein interactomes, and drug–target networks [113,114].

### 3.1. Proteinase Inhibitors

To date, three protease inhibitors have been approved by the Federal Drug Administration (FDA), bortezomib, carfilzomib, and ixazomib (Table 2). They are used to treat multiple myeloma [115,116,117]. Peptide borate inhibitor bortezomib blocks the chymotrypsin-like activity of the 26S proteasome to treat multiple myeloma patients [118,119,120,121]. Of note, use of bortezomib alone generally results in severe side-effects, which can be minimized through combining other drugs [122,123]. Bortezomib was also approved by FDA to treat mantle cell lymphoma [124]. Currently, use of bortezomib for other cancer types, such as autoimmune hemolysis and colon cancer, is under clinical trials [125,126].

Tetrapeptide epoxyketone carfilzomib selectively binds to the 26S proteasome to inhibit the activities of chymotrypsin-like protein, leading to accumulation of ubiquitinated substrates. In 2012, carfilzomib was approved by the FDA for clinical therapy of multiple myeloma [115,116,127,128,129]. Carfilzomib is currently at the stage of clinical trial for treatments of multiple cancers including neuroendocrine cancer, kidney cancer, lymphoma, acute myeloid, acute lymphoblastic leukemia, amyloidosis, and small-cell lung cancers [130].

Ixazomib (MLN2238, Ninlaro) is a boric acid derivative that inhibits chymotrypsin-like activity of 20S proteasome [130,131,132]. Compared with bortezomib, ixazomib displays a stronger antitumor effect [132]. Strikingly, treatment combining ixazomib and lenalidomide/dexamethasone can significantly increase the survival rate of myeloma patients [133]. Furthermore, ixazomib has completed a stage I clinical trial for glioblastoma, as well as a stage II clinical trial for malignant myeloid and lymphoid hematological cancers [130].

In addition to bortezomib, carfilzomib, and ixazomib, other protease inhibitors, such as marizomib, oprozomib, and delanzomib, are undergoing clinical trials for multiple myeloma and other cancers [115].

### 3.2. Drug Development Targeting the E1 Enzyme

Mammal genomes encode two E1s: UAE (also known as UBE1) and UBA6. Tak-243 (also known as MLN7243) is a small-molecule inhibitor specifically targeting UAE. Its application can reduce the overall mono- and polyubiquitination levels of cancer cells, leading to cancer cell death through an impaired cell-cycle process and accumulated DNA damage (Table 2) [134,135]. Tak-243 is currently undergoing clinical trials to treat refractory acute myeloid leukemia, refractory myelodysplastic syndrome, and chronic myelomonocytic leukemia [130].

### 3.3. Development of Drugs Targeting E3 Ligases

Generally, small0molecule drugs are designed using a lock-and-key mechanism to specifically target disease-related proteins. This strategy usually requires target proteins possessing a proper pocket region in structure as a binding site for small molecules. For those targets lacking unique pockets, two key strategies can be employed. First of all, proteolysis-targeting chimera (PROTAC) technology is a useful platform driving target proteins for degradation. The heterogeneous chimera is mainly composed of two important components, including one small-molecule ligand to specifically bind its target protein and one ligand to recruit an E3 to mediate ubiquitination of the captured protein for degradation [136,137,138,139]. Two examples designed using PROTAC with E3 cereblon (CRBN) as a ligand are Arv-110 and Arv-471 (Table 2) that are currently at the clinical trial stage. Arv-110 targets androgen receptor to treat metastatic castration-resistant prostate cancer (mCRPC), while Arv-471 targets estrogen receptor to treat breast cancer [130,138]. Although PROTAC technology is promising to develop drugs for human diseases, the sizes of molecules designed are always big. Moreover, choices of small-molecule ligands with high specificity in capturing E3 ligases are relatively limited. Another attractive strategy is represented by molecular glue degraders. They are small molecules that can modify molecular surface and, thus, enable novel interactions between targeted proteins and E3s, achieving ubiquitination-mediated protein degradation [140]. Compared with PROTAC, molecular glue degraders are smaller in size and easier to optimize their chemical properties. Several molecular glue degraders have been reported. CC-90009 recruits the CUL4–DDB1–CRBN–RBX1 (CRL4^CRBN^) E3 complex to ubiquitinate G1-to-S phase transition 1 (GSPT1) for proteasomal degradation. It is currently undergoing a phase I/II clinical trial to treat leukemia [130,141]. Serdemetan (JNJ-26854165) serves as an HDM2 E3 ligase antagonist, which mainly prevents degradation of p53 by inhibiting the interaction of HDM2 with p53. Moreover, serdemetan is capable of inhibiting cholesterol transport. It is under clinical investigation for human cell lymphoma and multiple myoma (Table 2) [130,142,143].

### 3.4. MDM2 Inhibitors

E3 MDM2 interacts with tumor suppressor p53 to mediate its ubiquitination and proteasome degradation. Targeting the MDM2–p53 complex is an effective modality for cancer treatment [79,144]. A panel of small-molecule inhibitors of MDM2, including RG7112, RG7388, AMG-232, APG-115, BI-907828, CGM097, sirmadlin, milademetan, SAR405838, MK-8242, PRIMA1, and APR-246, is undergoing clinical trials investigating their therapeutic effects on cancers (Table 2) [145,146,147].

### 3.5. IAPs Inhibitors

RING-type E3 IAPs are involved in cancer development regulation through inhibiting apoptosis and may serve as potential target for cancer immunotherapy [148,149,150]. GDC-0152, LCL161, AT-406, AEG40826, TL-32711, and APG-1387 are drugs targeting IAPs at the clinical stages (Table 2) [151]. These small-molecule drugs enable the interaction of XIAP and cIAP1 to undergo proteasome dependent degradation. Their design is inspired by the natural compound IAP antagonist Smac/DIABLO domain [152]. GDC-0152 is the first IAP antagonist that has been put into clinical trials [153], which is used to treat solid cancers via intravenous injection. As the first oral Smac mimic inhibitor of apoptosis [154], the effects of Lcl161 in treating solid tumors, including small-cell lung cancer, breast cancer, myelofibrosis, and thrombocytosis, are now under clinical investigation [130].

### 3.6. SCF^Skp2^ Inhibitors

As discussed above, Skp2 is overexpressed in various cancers and promotes tumor occurrence and malignancy. Skp2 has been suggested as a promising target for anticancer drug development [155,156,157,158]. Currently, a panel of drugs targeting Skp2 is under clinical trials for treatment of acute lymphoblastic leukemia, prostate cancer, prostate adenocarcinoma, and colon cancer. These promising drugs include curcumin, quercetin, lycopene, silibinin, epigalocatechin-3-gallate (EGCG), and vitamin D3 [130,159,160,161,162,163].

### 3.7. Nedd4-1 Inhibitors

Nedd4 family E3 Nedd4-1 is highly expressed in non-small-cell lung cancer (NSCLC), colon cancer, gastric cancer, and other cancers to promote cancer development [164,165]. To target Nedd4-1 is a potential avenue for the development of novel anticancer drugs [166,167,168]. Indole-3-carbinol (I3C) is derived from Cruciferae (Table 2). It possesses an indole carbinol structure and acts as an inhibitor against Nedd4-1, and it is currently at the clinical trial stage to treat human melanoma and breast cancer [130,169,170].

### 3.8. Hoil-1-Interacting Protein (HOIP) Inhibitors

RBR-type E3 HOIP is involved in regulating human immune response [171,172]. Bendamustine (Table 2), as an effective drug inhibiting HOIP, has been approved by the FDA and the European Medicines Agency (EMA) for the treatment of multiple myeloma, chronic leukemia, rituximab/refractory follicular, and low-grade lymphoma [173]. Furthermore, it is undergoing clinical trials for other human cancers, including ovarian cancer, metastatic HER2-negative breast cancer, and different types of lymphoma [130,174,175,176].

**Table 2 ijms-23-15104-t002:** Drugs targeting ubiquitination factors approved or in clinical trials.

Drugs	Mechanism and Application	References
Arv-110	As an inhibitor of PROTACs with E3 CRBN as a ligand, Arv-110 targets the androgen receptor to treat mCRPC.	[130,138]
Arv-471	As an inhibitor of PROTACs with E3 CRBN as a ligand, Arv-471 targets the estrogen receptor to treat breast cancer.	[130,138]
Bortezomib	1. Bortezomib blocks the chymotrypsin-like activity of the 26S proteasome to treat multiple myeloma patients. 2. Bortezomib is the first drug approved by the FDA for the treatment of multiple myeloma and myeloma.	[118,119,120,121]
Bendamustine	1. Bendamustine inhibits RBR-type E3 HOIP. 2. Bendamustine was approved by the FDA for clinical therapy of EMA for the treatment of multiple myeloma, chronic leukemia, rituximab/refractory follicular, and low-grade lymphoma.	[130,174,175,176]
Carfilzomib	1. Carfilzomib selectively binds to the 26S proteasome to inhibit the activities of chymotrypsin-like protein, leading to the accumulation of ubiquitinated substrates. 2. Carfilzomib was approved by the FDA for clinical therapy of multiple myeloma.	[115,116,127,128,129]
CC-90009	CC-90009 recruits the CRL4^CRBN^ E3 complex to ubiquitinate GSPT1 for proteasomal degradation.	[130,141]
Curcumin, quercetin, lycopene, silibinin, EGCG, and vitaminD3	These drugs promote apoptosis by downregulating Skp2 levels.	[130,159,160,161,162,163]
GDC-0152, LCL161, AT-406, AEG40826, TL-32711, and APG-1387	The drugs bind to the BIR3 domain of the IAP proteins through its N terminus, blocking caspase–IAP complex formation and promoting apoptosis.	[150,151,152]
Ixazomib	1. Ixazomib belongs to boric acid and inhibits chymotrypsin-like activity of 20S proteasome. 2. Ixazomib was approved by the FDA for clinical therapy of multiple myeloma.	[130,131,132]
I3C	I3C binds to the HECT domain of Nedd4-1 and inhibits cancer cell proliferation.	[130,169,170]
RG7112, RG7388, AMG-232, APG-115, BI-907828, CGM097, sirmadlin, milademetan, SAR405838, MK-8242, PRIMA1, and APR-246	These drugs inhibit the interaction of MDM2 with p53 so as to maintain p53 activity.	[79,144,145]
Serdemetan (JNJ-26854165)	1. Serdemetan prevents degradation of p53 by inhibiting the interaction of HDM2 with p53. 2. Serdemetan is capable of inhibiting cholesterol transport.	[130,142,143]
Tak-243 (MLN7243)	Tak-243 is the first UAE inhibitor based on a ubiquitination mechanism to enter the clinic.	[134,135]

## 4. Concluding Remarks and Perspectives

Ubiquitination-related factors are highly involved in regulating cancer development, providing them as promising “druggable” targets for cancer therapy. In spite of striking expectations, inhibitors or agonists against E1/E2/E3 components might possibly result in adverse effects on normal activities in cells since ubiquitination extensively monitors diverse fundamental biological processes. Thus, achieving high specificity in cancer cell targeting is critical to meet this challenge. Innovative solutions could be derived from multidisciplinary integration. For example, pH-response nanomaterials can serve as vectors mediating unique release of inhibitors/agonists against E1/E2/E3 in the cancer-specific pH environment. Furthermore, it will be of great importance to dissect the detailed mechanisms underlying the ubiquitination-related regulation of cancer onset and development. To date, increasing insights and techniques have been developed that would open a new arena for cancer therapy in the future.

## Figures and Tables

**Figure 1 ijms-23-15104-f001:**
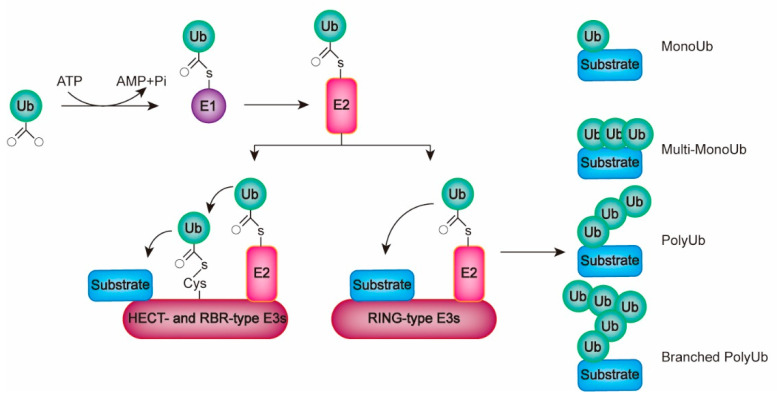
The ubiquitination machinery. Ubiquitination is achieved through the sequential activity of E1, E2, and E3. RING-type E3s can mediate a direct ligation of Ub from E2s to substrates, whereas HECT-/RBR-type E3s require two-step reactions. Typical types of Ub linkages include MonoUb, multi-MonoUb, poly-Ub, and branched poly-Ub.

## Data Availability

No applicable.

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
