# Peer review of "Progress in Anticancer Drug Development Targeting Ubiquitination-Related Factors"

_ijms, 2022, doi:10.3390/ijms232315104_

Round 1
Reviewer 1 Report
In this manuscript, the authors focus on the relationship between ubiquitination and cancer, including detailed descriptions of the mechanisms of ubiquitination, cancer-related ubiquitination factors and drug developments targeting ubiquitination-related factors. This manuscript is detailed but there are still some issues which need to be addressed and major revision should be made before acceptance.
Mechanisms of ubiquitination
(1) P1-P2, L42-L83, since the ubiquitin process involves a variety of enzymes, it is suggested to supplement the schematic diagram to illustrate it.
(2) P1, L26-L27, the expression of “The methionine-1 (M1) and all K residues in Ub, including K6, K11, K27, 26 K29, K33, K48 and K63, can serve as a linkage site” is not rigorous since it is mentioned in the literature “The Role of E3s in Regulating Pluripotency of Embryonic Stem Cells and Induced Pluripotent Stem Cells” that C/S/T/Y can also serve as a ubiquitin site in substrate proteins except M1. (Original Text, “However, some non-K residues of proteins, such as methionine-1 (M1), cysteine (C), serine (S), threonine (T) and tyrosine (Y), can serve as ubiquitination sites under some particular conditions”)
(3) P1, L37, the expression of “the same Ub linkage” is not rigorous because although this linkage includes the same Ub molecule, it also includes diverse residue connection sites as mentioned above.
(4) P1, L25, it is suggested that G76 be explained because it appeared for the first time.
Cancer-related ubiquitination factors
(1) The unclear classification of cancer-associated ubiquitination factors by different mechanisms leads to structural confusion, so please consider whether to rearrange this section in the order of E1/E2/E3.
Drug development targeting ubiquitination-related factors
(1) It is suggested that tables be added to aid in the description of the various antitumor agents that target ubiquitination-related factors.
(2) Items 3.4 - 3.8 are proposed to be incorporated under 3.3.
Spelling errors
L43, "produce" should be used in the singular form.
L52, "interaction" should be used in plural form.
L61, "contains" should be used in plural form.
L79, the parallel word such as "and" should be added in the sentence of "DNA replication[34, 35], transcription[36], inflammatory response".
L110, position of one more character in brackets.
L128, one more character position between "dNTPs to".
L178, "development" should be used in plural form.
L181-L182, the sentence of "These drugs could be derived form small molecules" has grammatical errors.
L194-L195, the sentence of "To date, use of Bortezomb other cancer types, such as autoimmune hemolysis and colon cancer, is under clinical trials " has grammatical errors.
L199-L200, "treatment" should be used in plural form.
L230-L231, the sentence of " Two examples designed using PROTAC are Arv-110 and arv-471 that currently under clinical trial stage" has grammatical errors.
L248, the word of "targetted" is misspelled.
L250-L251, there is an extra period in the sentence of " Targeting MDM2-p53 complex is an effective modality for cancer 250 treatment[65, 144].. ".
L253, "effect" should be used in plural form.
L257, the sentence of "the apoptosis pathway is blocked[87]" lacks a period.
L257-L258, the sentence of "the apoptosis pathway is blocked[87] RING-type E3 IAPs inhibit apoptosis by binding to caspase family members[148, 149] IAPs show great potential in cancer immunotherapy, [Michie j et al., 2020]" does not make sense.
L260, "stage" should be used in plural form.
L262-L264, the sentence of "the design of these small drug molecules is inspired by the natural compound IAP antagonist Smac/DIABLO domain[151], gdc-0152 is the first IAPs antagonist that has been put into clinical trials[152]" has grammatical errors.
L265, "effect" should be used in plural form.
L272, "treatment" should be used in plural form.
L277, "development" should be used in plural form.
L279-L281, the sentence of "It possesses indole carbinol structure and acts as a inhibitor against nedd4-1, and currently under clinical trial stage to treat human melanoma and breast cancer" has grammatical errors and "a" should be modified to "an".
L288, one more character position between "breast cancer and different types".
Author Response
Response to Reviewer 1 Comments
In this manuscript, the authors focus on the relationship between ubiquitination and cancer, including detailed descriptions of the mechanisms of ubiquitination, cancer-related ubiquitination factors and drug developments targeting ubiquitination-related factors. This manuscript is detailed but there are still some issues which need to be addressed and major revision should be made before acceptance.
Mechanisms of ubiquitination
- P1-P2, L42-L83, since the ubiquitin process involves a variety of enzymes, it is suggested to supplement the schematic diagram to illustrate it.
Response: We have provided it in the revised manuscript.
- P1, L26-L27, the expression of “The methionine-1 (M1) and all K residues in Ub, including K6, K11, K27, 26 K29, K33, K48 and K63, can serve as a linkage site” is not rigorous since it is mentioned in the literature “The Role of E3s in Regulating Pluripotency of Embryonic Stem Cells and Induced Pluripotent Stem Cells” that C/S/T/Y can also serve as a ubiquitin site in substrate proteins except M1. (Original Text, “However, some non-K residues of proteins, such as methionine-1 (M1), cysteine (C), serine (S), threonine (T) and tyrosine (Y), can serve as ubiquitination sites under some particular conditions”)
Response: We agree with the reviewer's comment. In the revised manuscript, the sentence has been changed and replaced with "All K residues in Ub, including K6, K11, K27, 26 K29, K33, K48 and K63, can serve as a linkage site. Of note, in some particular cases, Ub can be conjugated to non-K residues, such as methionine-1 (M1), cysteine (C), serine (S), threonine (T) and tyrosine (Y), can serve as ubiquitination sites under some particular conditions".
- P1, L37, the expression of “the same Ub linkage” is not rigorous because although this linkage includes the same Ub molecule, it also includes diverse residue connection sites as mentioned above.
Response: The focus of this information mainly is about the interesting finding that the same type of Ub chain could result in distinct effect on substrate proteins.
- P1, L25, it is suggested that G76 be explained because it appeared for the first time.
Response: We have fixed it in the revised manuscript.
Cancer-related ubiquitination factors
- The unclear classification of cancer-associated ubiquitination factors by different mechanisms leads to structural confusion, so please consider whether to rearrange this section in the order of E1/E2/E3.
Response: We thank the reviewer for the insightful suggestion, and have rearranged this section with the order of E1/E2/E3 in the revised manuscript.
Drug development targeting ubiquitination-related factors
- It is suggested that tables be added to aid in the description of the various antitumor agents that target ubiquitination-related factors.
Response: We have provided the table in the revised manuscript.
(2) Items 3.4 - 3.8 are proposed to be incorporated under 3.3.
Response: We thank the reviewer for the insightful suggestion, and have incorporated them under 3.3 in the revised manuscript.
Spelling errors
L43, "produce" should be used in the singular form.
Response: We have corrected it in the revised manuscript.
L52, "interaction" should be used in plural form.
Response: We have corrected it in the revised manuscript.
L61, "contains" should be used in plural form.
Response: We have corrected it in the revised manuscript.
L79, the parallel word such as "and" should be added in the sentence of "DNA replication[34, 35], transcription[36], inflammatory response".
Response: We have added it in the revised manuscript.
L110, position of one more character in brackets.
Response: We have corrected it in the revised manuscript.
L128, one more character position between "dNTPs to".
Response: We have removed it in the revised manuscript.
L178, "development" should be used in plural form.
Response: We have corrected it in the revised manuscript.
L181-L182, the sentence of "These drugs could be derived form small molecules" has grammatical errors.
Response: The "form" has been changed to "from "in the revised manuscript.
L194-L195, the sentence of "To date, use of Bortezomb other cancer types, such as autoimmune hemolysis and colon cancer, is under clinical trials " has grammatical errors.
Response: It has been changed to "To date, use of Bortezomb for other cancer types, such as autoimmune hemolysis and colon cancer, is under clinical trials" in the revised manuscript.
L199-L200, "treatment" should be used in plural form.
Response: We have corrected it in the revised manuscript.
L230-L231, the sentence of " Two examples designed using PROTAC are Arv-110 and arv-471 that currently under clinical trial stage" has grammatical errors.
Response: It has been changed to "Two examples designed using PROTAC are Arv-110 and arv-471 that are currently under clinical trial stage" in the revised manuscript.
L248, the word of "targetted" is misspelled.
Response: It has been changed as "targeted" in the revised manuscript.
L250-L251, there is an extra period in the sentence of " Targeting MDM2-p53 complex is an effective modality for cancer 250 treatment[65, 144].. ".
Response: We have corrected it in the revised manuscript.
L253, "effect" should be used in plural form.
Response: We have corrected it in the revised manuscript.
L257, the sentence of "the apoptosis pathway is blocked[87]" lacks a period.
Response: We have added it in the revised manuscript.
L257-L258, the sentence of "the apoptosis pathway is blocked[87] RING-type E3 IAPs inhibit apoptosis by binding to caspase family members[148, 149] IAPs show great potential in cancer immunotherapy, [Michie j et al., 2020]" does not make sense.
Response: "One of the major reasons is that the apoptosis pathway is blocked[87] RING-type E3 IAPs inhibit apoptosis by binding to caspase family members[148, 149] IAPs show great potential in cancer immunotherapy, " in the original manuscript has been revised as "RING-type E3 IAPs is involved in cancer development regulation through inhibiting apoptosis, and may serve as potential target for cancer immunotherapy [148, 149,Michie j et al., 2020] " in the revised manuscript.
L260, "stage" should be used in plural form.
Response: We have corrected it in the revised manuscript.
L262-L264, the sentence of "the design of these small drug molecules is inspired by the natural compound IAP antagonist Smac/DIABLO domain[151], gdc-0152 is the first IAPs antagonist that has been put into clinical trials[152]" has grammatical errors.
Response: In the revised manuscript, the original sentences "These small drug molecules enable the interaction of X-IAP and CIAP1 to undergo proteasome dependent degradation, and the design of these small drug molecules is inspired by the natural compound IAP antagonist Smac/DIABLO domain[151], gdc-0152 is the first IAPs antagonist that has been put into clinical trials[152], which is used to treat solid cancers by intravenous injection. " have been changed to "These small drug molecules enable the interaction of X-IAP and CIAP1 to undergo proteasome dependent degradation. The design of these small drug molecules is inspired by the natural compound IAP antagonist Smac/DIABLO domain[151], and gdc-0152 is the first IAPs antagonist that has been put into clinical trials[152], which is used to treat solid cancers by intravenous injection. "
L265, "effect" should be used in plural form.
Response: We have corrected it in the revised manuscript.
L272, "treatment" should be used in plural form.
Response: We have corrected it in the revised manuscript.
L277, "development" should be used in plural form.
Response: We have corrected it in the revised manuscript.
L279-L281, the sentence of "It possesses indole carbinol structure and acts as a inhibitor against nedd4-1, and currently under clinical trial stage to treat human melanoma and breast cancer" has grammatical errors and "a" should be modified to "an".
Response: We have corrected it in the revised manuscript.
L288, one more character position between "breast cancer and different types".
Response: We have removed the additional character position in the revised manuscript.
Reviewer 2 Report
In this manuscript, the authors reviewed the progress in anti-cancer drug development targeting ubiquitination-related factors. This is an informative review and will benefit the readers who seek for information on this subject.
Please find my comments below:
- A figure illustrating/summarizing the mechanisms of ubiquitination described in the first paragraph would be useful for the reader and increase the value of the article
- Adding two tables summarizing the cancer-related ubiquitination factors and the drugs targeting such factors would be extremely useful for the reader
- I would expect a paragraph discussing the future perspectives and possible limitations of agents targeting ubiquitination-related factors.
- A “conclusions” sections should be added
Minor
Line 79: I guess [Spence J, 2000] should be deleted.
Author Response
Response to Reviewer 2 Comments
In this manuscript, the authors reviewed the progress in anti-cancer drug development targeting ubiquitination-related factors. This is an informative review and will benefit the readers who seek for information on this subject.
Please find my comments below:
- A figure illustrating/summarizing the mechanisms of ubiquitination described in the first paragraph would be useful for the reader and increase the value of the article
Response: We have provided it in the revised manuscript.
- Adding two tables summarizing the cancer-related ubiquitination factors and the drugs targeting such factors would be extremely useful for the reader
Response: We have provided it in the revised manuscript.
- I would expect a paragraph discussing the future perspectives and possible limitations of agents targeting ubiquitination-related factors.
Response: We have provided it in the revised manuscript.
Minor
Line 79: I guess [Spence J, 2000] should be deleted.
Response: We have fixed it in the revised manuscript.
Reviewer 3 Report
Ever since the discovery of ubiquitin (Ub) four decades ago this small protein has been linked to multiple cellular processes, including cell proliferation, development, immune responses, and numerous human diseases including cancer. Development of effective cancer therapeutic strategies relies on our ability to interfere with cellular processes that are dysregulated in tumors. Given the essential role of the ubiquitin proteasome system (UPS) in regulating a myriad of cellular processes, it is not surprising that malfunction of UPS components is implicated in numerous human diseases, including many types of cancer.
The authors summarize progress in anti-cancer drug development targeting ubiquitination-related factor. The work is divided into three parts. In the first part, the authors describe the mechanisms of ubiquitination. In the second part authors describe the Cancer-related ubiquitination factors. Finally the authors describe drug development targeting ubiquitination-related factors.
The review is clearly written and relevant to the field. However, attention is drawn to a large number of abbreviations, which are not always developed, which makes it difficult to read the study.
The topics discussed by the authors are very popular, as evidenced by over a dozen items in review articles from the last five years. However, the presented overview seems to be up-to-date and interesting for the scientific community.
The work contains a large number of 175 references, mainly from the latest publications. I did not notice a large number of self-citations in the presented work.
In the publication, I miss a summary of the presented information in conclusion. The authors could present conclusions as to which mechanisms and directions of research development seem to be the most promising.
Author Response
Response to Reviewer 3 Comments
Ever since the discovery of ubiquitin (Ub) four decades ago this small protein has been linked to multiple cellular processes, including cell proliferation, development, immune responses, and numerous human diseases including cancer. Development of effective cancer therapeutic strategies relies on our ability to interfere with cellular processes that are dysregulated in tumors. Given the essential role of the ubiquitin proteasome system (UPS) in regulating a myriad of cellular processes, it is not surprising that malfunction of UPS components is implicated in numerous human diseases, including many types of cancer.
The authors summarize progress in anti-cancer drug development targeting ubiquitination-related factor. The work is divided into three parts. In the first part, the authors describe the mechanisms of ubiquitination. In the second part authors describe the Cancer-related ubiquitination factors. Finally the authors describe drug development targeting ubiquitination-related factors.
The review is clearly written and relevant to the field. However, attention is drawn to a large number of abbreviations, which are not always developed, which makes it difficult to read the study.
Response: We have provided it in the revised manuscript.
The topics discussed by the authors are very popular, as evidenced by over a dozen items in review articles from the last five years. However, the presented overview seems to be up-to-date and interesting for the scientific community.
The work contains a large number of 175 references, mainly from the latest publications. I did not notice a large number of self-citations in the presented work.
In the publication, I miss a summary of the presented information in conclusion. The authors could present conclusions as to which mechanisms and directions of research development seem to be the most promising.
Response: We have provided it in the revised manuscript.
Round 2
Reviewer 1 Report
The sentence of "Of note, in some particular cases, Ub can be conjugated to non-K residues, such as methionine-1 (M1), cysteine (C), serine (S), threonine (T) and tyrosine(Y), can serve as ubiquitination sites under some particular conditions[1-4]." has grammatical errors.
Author Response
The sentence of "Of note, in some particular cases, Ub can be conjugated to non-K residues, such as methionine-1 (M1), cysteine (C), serine (S), threonine (T) and tyrosine(Y), can serve as ubiquitination sites under some particular conditions[1-4]." has grammatical errors.
Response: We have corrected it in the revised manuscript. This sentence has been changed to "Of note, in some particular cases, Ub can be conjugated to non-K residues, such as methionine-1 (M1), cysteine (C), serine (S), threonine (T) and tyrosine(Y),[1-4]."
Reviewer 2 Report
The authors addressed all my comments
Author Response
Thank you!